# Antibiotic treatment inhibits paclitaxel chemotherapy-induced activity deficits in female mice

**Corena V. Grant** [1], **Kelley Jordan**[1], **Melina M. Seng**[1], **Leah M. Pyter** [1,2]*

**1** Institute for Behavioral Medicine Research, Ohio State University, Columbus, Ohio, United States of America, **2** Department of Psychiatry and Behavioral Health, Ohio State University, Columbus, Ohio, United States of America

* leah.pyter@osumc.edu

**Data Availability Statement:** All data used in the generation of figures is available at https://doi.org/10.6084/m9.figshare.20418951.

**Funding:** This work was supported by National Institute of Health, National Cancer Institute grant

## Abstract

Chemotherapy, a mainstay in the treatment of cancer, is associated with severe and debilitating side effects. Side effects can be physical (e.g., gastrointestinal distress, anemia, and hair loss) or mental (e.g., fatigue, cognitive dysfunction). Chemotherapy is known to alter the gut microbiota; thus, communication through the gut-brain axis may influence behavioral side effects. Here, we used a clinically-relevant paclitaxel chemotherapy regimen in combination with antibiotics to test the hypothesis that gut microbes contribute to chemotherapy-associated fatigue-like behaviors in female mice. Data presented suggest that chemotherapy-altered gut microbes contribute to fatigue-like behaviors in mice by disrupting energy homeostasis.

## Introduction

Despite the associated side effects, chemotherapy remains a mainstay in the treatment of many cancers. In fact, 9.8 million of 17 million new cancer cases in 2018 required chemotherapy treatment, a number that is only expected to rise [1]. Prominent and debilitating side effects of chemotherapy include gastrointestinal distress and brain-mediated behavioral consequences. At least 70% of patients in one study experienced gastrointestinal side effects including constipation, diarrhea, and mucositis, with fatigue as the most common behavioral side effect by far [2]. Cancer-related fatigue (CRF) occurs in approximately 80% of all patients receiving chemotherapy and can persist for years after treatment cessation [2]. CRF is different from colloquial fatigue in that it is not only peripherally-mediated (i.e., muscle impairment), but is also centrally-mediated and does not improve with rest [3]. Indeed, chemotherapy-associated fatigue-like behaviors measured in mouse models using wheel-running, home cage locomotion, and activity in an open field appear to be centrally-mediated rather than a result of impaired motor function [4]. The profound impact of fatigue reduces quality of life and increases treatment noncompliance, thereby increasing mortality [5, 6]. Unfortunately, the development of CRF remains poorly understood and treated. Therefore, the need to identify mechanisms of CRF and targets of intervention are critical for improving survival in the ever-increasing population of cancer patients and survivors.

CA216290 and an associated personnel supplement -04S2 (L.P., C.G. [trainee]). The funders had no role in study design, data collection and analysis, decision to publish, or preparation of the manuscript.

**Competing interests:** The authors have declared that no competing interests exist.

The melanocortin system plays a critical role in energy homeostasis relevant to fatigue. Under normal physiological conditions, pro-opiomelanocortin (POMC) neurons are activated by leptin, which causes the translation of the *Pomc* gene and post-translational processing leads to the secretion of α-melanocyte-stimulating hormone (MSH), among other peptides. A-MSH inhibits neurons in the paraventricular nucleus of the hypothalamus to increase energy expenditure and drive satiety [7]. Transcription of *Pomc*, and associated production of α-MSH, can also be stimulated by serotonin (5-HT) binding to its receptor, 5-HTR2C, located on POMC neurons [8]. Ghrelin, a hormone secreted by the stomach, opposes leptin, and stimulates feeding behavior by activating its receptor, growth hormone secretagogue receptor (GHSR), on AgRP/NPY neurons in the arcuate nucleus of the hypothalamus [7]. Additional neuropeptides, orexins, also regulate energy balance. Orexin receptors, namely OX1R (*Hcrtr1* gene), are expressed on POMC neurons; activation of these receptors inhibits the activity of these neurons, thereby inhibiting energy expenditure and driving feeding behavior [9, 10]. It has been demonstrated that chemotherapy can suppress hypothalamic orexin activity, leading to chemotherapy-associated fatigue in rodent models [11].

The ability of the gut microbiome to modulate energy balance and fatigue via the melanocortin axis has been demonstrated. For example, Schéle et al. observed decreases in gene expression of energy-regulating genes in the hypothalamus of conventionally-raised mice in comparison to germ-free mice [12]. Additionally, conventionally-raised mice did not lose weight in response to leptin injection [12]. The gut bacteria-derived protein caseinolytic peptidase B protein homolog is a mimetic of α-MSH and plays a role in the pathogenesis of eating disorders [13], and chronic fatigue syndrome patients have significantly lower gut bacterial phylogenetic diversity than healthy patients [14]. Studies from our lab and others have identified the gut microbiota as a regulator of paclitaxel chemotherapy side effects in rodents and in human patients [15–17]. We have demonstrated that a gut microbial transplant from paclitaxel-treated donor mice induced similar neuroinflammation and behavioral side effects in germ-free mice as mice treated with paclitaxel directly, suggesting paclitaxel-altered microbiota is a major driver of paclitaxel side effect development [15].

Here, we aimed to determine the extent to which gut microbial reduction prevents chemotherapy-associated locomotor suppression (i.e., fatigue) in mice. We hypothesized that treatment with broad-spectrum antibiotics would deplete microbes that contribute to chemotherapy side effects, and thereby attenuate locomotor deficits in association with dysregulation of the melanocortin axis. We treated mice with a clinically-relevant paclitaxel chemotherapy treatment regimen and assessed locomotor activity via multiple methods: voluntary wheel running and locomotion in an open field chamber. This study suggests that antibiotics may attenuate chemotherapy-associated lethargy during treatment. This intimates that the gut microbiome may play a role in the development of chemotherapy-associated fatigue potentially via modulation of energy homeostasis mechanisms.

## Results

### Antibiotic chow does not impact paclitaxel-associated weight loss

Prior to the start of treatment, mice were acclimated to running wheels and given antibiotic or control chow. As expected, this antibiotic cocktail in chow caused a significant decrease in fecal 16S rRNA as verified in vehicle-treated mice (**S1 Fig**). During the acclimation period, antibiotics caused a reduction in body mass that recovered to the level of control-chow mice by the time treatment started (**Fig 1B**, Chow effect day -8 to -4, $F_{1,26} = 6.95$, p<0.05, Chow x Day effect day -8 to -4, $F_{1,26} = 9.63$, p<0.01). This is likely driven by lower food intake in antibiotic-treated mice during this time period, which increases over baseline to reach the level of

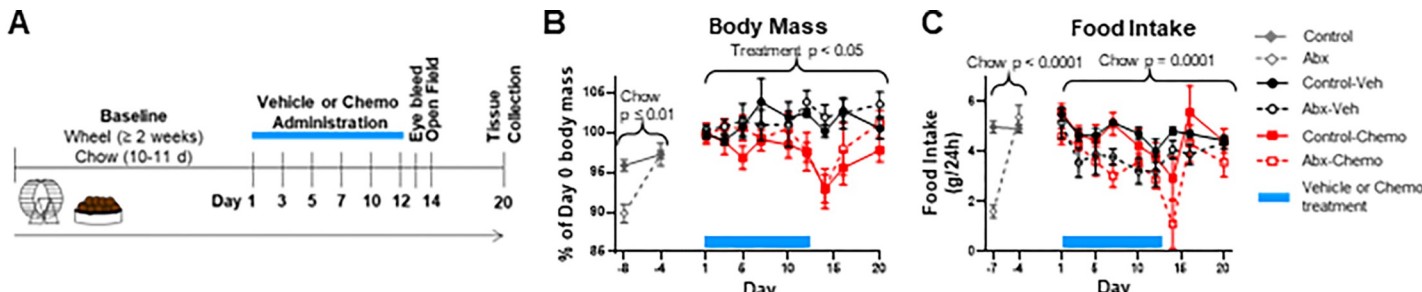

**Fig 1. Study design and the effect of treatment on body mass and food intake. A)** An outline of the study design including baseline exposure to experimental chow and running wheels, treatment with vehicle or paclitaxel, and timing of behavior and tissue collections **B)** Body mass represented as the % of day 0 (day prior to the start of vehicle or paclitaxel treatment) **C)** The average food intake per mouse per 24-hour period. Each point reflects the previous 48 hr. n = 5–8. p-values are representative of ANOVA main effects over the indicated time period.

control chow ([Fig 1C], Chow effect day -7 to -4, $F_{1,54} = 22.65$, p<0.0001, Chow x Day effect day -7 to -4, $F_{1,54} = 38.99$, p<0.0001). During the treatment and recovery period, chemotherapy caused mice to lose weight; vehicle did not ([Fig 1B], Treatment effect day 1 to 20, $F_{1,24} = 7.7$, p<0.05). Paclitaxel-associated weight loss was not associated with an overall reduction in food intake, though depending on the day, paclitaxel did reduce food intake ([Fig 1C], Treatment x Day effect day 1 to 20, $F_{8,162} = 2.23$, p<0.05). In contrast, antibiotics did not affect body mass, but reduced food intake over the treatment and recovery period ([Fig 1C], Chow effect day 1 to 20, $F_{1,26} = 20.76$, p = 0.0001). By the end of the paclitaxel recovery period, total fecal bacterial DNA (16S rRNA) returned to pre-antibiotic levels despite continuous antibiotic treatment ([S1 Fig]).

## Antibiotic chow attenuates paclitaxel-induced declines in activity

Wheel running activity was assessed during the dark phase as mice are most active during this phase as compared to the light-on phase of their cycle. Paclitaxel decreased dark phase revolutions (as % baseline) over the entire treatment period ([Fig 2A], Treatment effect, $F_{1,25} = 16.8$, p<0.001). Antibiotics interacted with paclitaxel treatment ([Fig 2A], Chow x Treatment effect, $F_{1,25} = 5.4$, p<0.05), driven by decreased activity in antibiotic versus control chow-fed mice in within the vehicle treatment group, but increased activity in antibiotic versus control chow-fed mice within the paclitaxel treatment group. Two refined wheel running measures were selected for focused visualization: the days of paclitaxel administration collapsed ([Fig 2A] middle) and the day of open field behavior testing ([Fig 2A] right) because open field behavior was assessed during the early part of this phase. During both periods, it is evident that paclitaxel decreased wheel running, and antibiotics affected wheel running dependent on vehicle or paclitaxel treatment ([Fig 2A], Treatment effects $F_{1,25} = 15.6$, p<0.001 and $F_{1,25} = 22.5$, p<0.0001, respectively; Chow x Treatment effects $F_{1,25} = 4.6$, p<0.05 and $F_{1,25} = 6.2$, p<0.05, respectively). Similar to running-wheel revolutions, in an open field, the antibiotic by paclitaxel interaction was again observed over the 15-min exploratory period ([Fig 2B], Chow x Treatment $F_{1,26} = 5.7$, p<0.05 and **2C** Chow x Treatment interaction $F_{1,25} = 4.6$, p<0.05), with antibiotics specifically attenuating paclitaxel-induced reductions in locomotion.

## Paclitaxel and antibiotics modulate energy-regulating systems

Ghrelin and leptin levels were measured in the same mice 1 and 8 days after the final dose of paclitaxel. One day after treatment completion, a trend for paclitaxel to increase ghrelin was observed; this likely did not reach significance because of high variability in the Control-Chemo group ([Fig 3A], Treatment effect $F_{1,21} = 4.0$, p = 0.06). Coincidentally, an antibiotic by

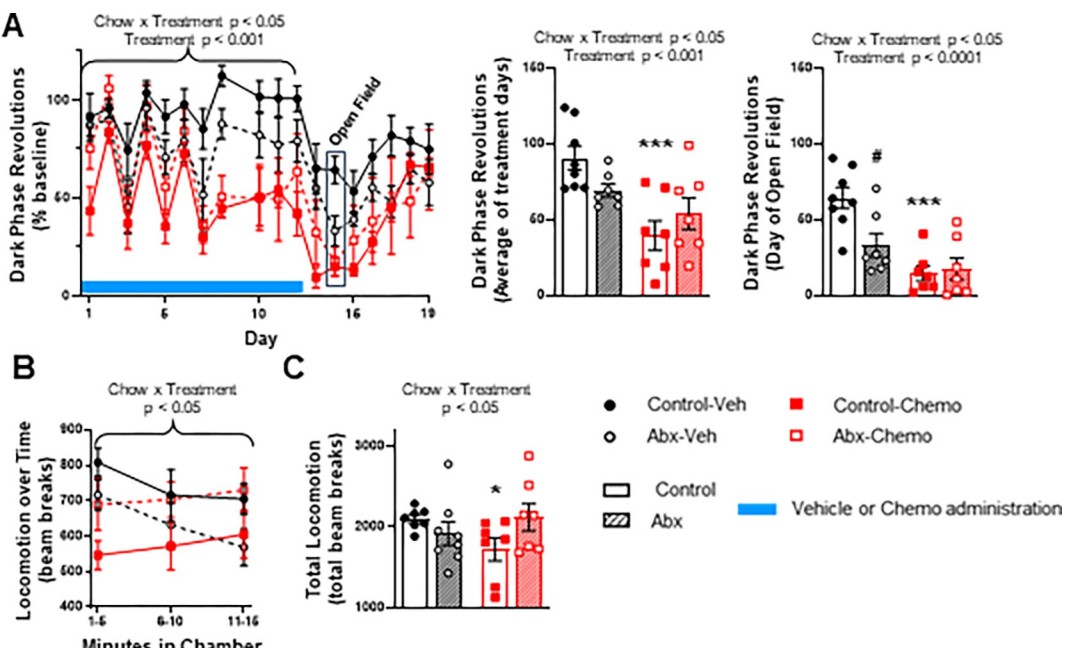

**Fig 2. Effect of paclitaxel and antibiotics on measures of fatigue. A)** The dark phase revolutions represented as a fraction of the average of the baseline revolutions over the entire experiment. The average revolutions on treatment days and on the day of open field are highlighted in separate bar graphs. **B)** Total locomotion represented as beam breaks in an open field chamber over a 15 min time period, each point represents 5 min bins of time **C)** Compiled total beam breaks for the entire 15 min of activity in the open field chamber. n = 6–8. * represents significant post-hoc test between treatment groups (e.g., Control-Veh vs. Control-Chemo); # represents significant post-hoc test between chow groups (e.g., Control-Veh vs. Abx-Veh). p-values above brackets are ANOVA main effect results.

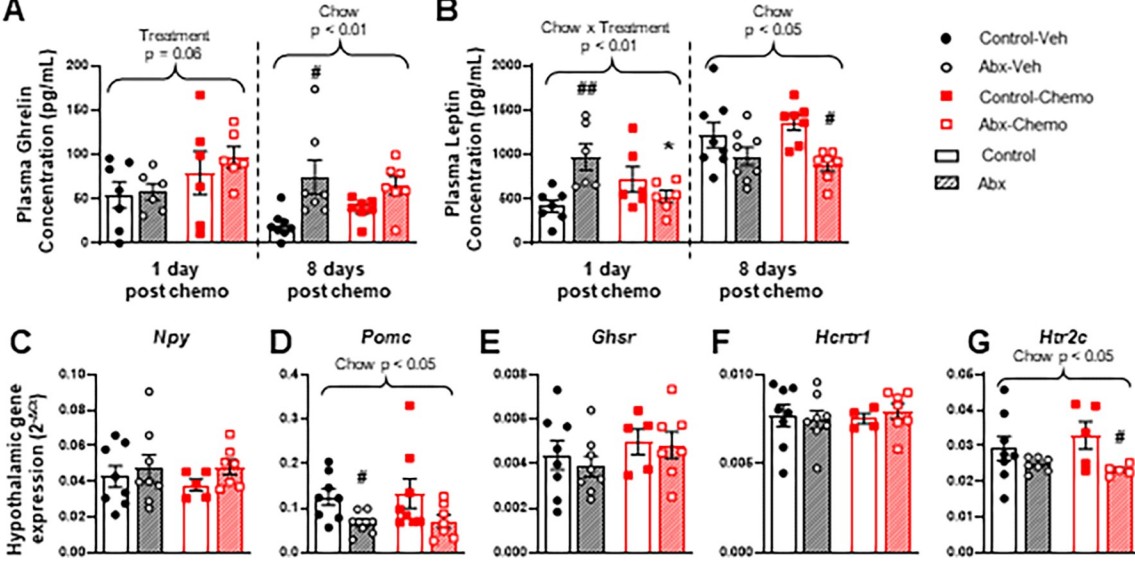

**Fig 3. The effect of paclitaxel and antibiotics on plasma feeding hormones and hypothalamic melanocortin pathway gene expression.** Plasma **A)** ghrelin and **B)** leptin concentration measured 1 day and 8 days after the final vehicle or paclitaxel treatment. Hypothalamic gene expression of **C)** *Npy* **D)** *Pomc* **E)** *Ghsr* **F)** *Hcrtr1* and **G)** *Htr2c* in tissue collected 8 days after the final vehicle or paclitaxel treatment. n = 5–8. * represents significant post-hoc test between treatment groups (e.g., Control-Veh vs. Control-Chemo); # represents significant post-hoc test between chow groups (e.g., Control-Veh vs. Abx-Veh).). p-values above brackets are ANOVA main effect results.

paclitaxel interaction on leptin concentrations was observed and driven primarily by antibiotics increasing the concentration in vehicle mice; among antibiotic-treated mice, paclitaxel decreased leptin concentrations compared to vehicle (**Fig 3B**, Chow x Treatment effect $F_{1,21}$ = 11.33, p = 0.003). Eight days following the final paclitaxel dose, antibiotics increased ghrelin in circulation (**Fig 3A**, Chow effect $F_{1,25}$ = 12.7, p<0.01); mice were still eating chow containing antibiotics at this time. Consistent with this, antibiotic chow decreased the plasma concentration of leptin when measured 8 days after the last treatment (**Fig 3B**, Chow effect $F_{1,26}$ = 11.5, p<0.01). Melanocortin pathway-related gene expression was investigated in the hypothalamus 8 days after the last treatment. The expression of *Npy*, *Pomc*, *Ghsr*, *Hcrtr1*, and *Htr2c* were not changed by paclitaxel treatment, however, *Pomc* and *Htr2c* expression was decreased by antibiotic chow (**Fig 3D**, Chow effect $F_{1,26}$ = 7.7, p<0.05 and **Fig 3G**, Chow effect $F_{1,21}$ = 6.6, p<0.05).

## Discussion

This study aimed to identify the extent to which, and how, gut microbiota-targeted antibiotics could alter the development of paclitaxel chemotherapy-associated fatigue. This work is particularly urgent as chemotherapy is a mainstay in the treatment of many cancers and is associated with side effects that lead to increased patient mortality, including fatigue. Here we demonstrate that antibiotics could alleviate some chemotherapy-associated fatigue potentially via coincident disruption to energy homeostasis mechanisms.

As previously published, this paclitaxel chemotherapy regimen transiently reduced body mass compared to vehicle-treated mice [4, 15], which was not due to a difference in food intake. However, antibiotic-treated mice ate significantly less chow than control mice without an impact on body mass. It is probable that this is due to the aversive taste of antibiotics because the reduction is observed throughout the entire experiment, and others have shown that mice reduce water intake when the water is formulated with a similar antibiotic cocktail containing ampicillin, vancomycin, neomycin, and metronidazole [18]. Interestingly, during baseline acclimation of food and running wheels, mice initially consumed less antibiotic-containing food, however by the time of vehicle or paclitaxel treatment, that was not the case. During the course of treatment, though, mice again consumed less antibiotic-containing chow, however weight loss was only impacted by chemotherapy treatment, not a decrease in antibiotic food intake. This is likely due to the metabolic changes documented in patients treated with chemotherapy [19]. Further, the increase in ghrelin and decrease in leptin caused by antibiotics (measured 8 days post treatment) would suggest food intake decreases are occurring despite appetite-stimulating signals. Despite the reduced food intake, we have previously demonstrated that this antibiotic chow is effective at decreasing gut bacterial alpha diversity [15]. Further demonstrated here is a recovery of the levels of fecal bacterial DNA (16S rRNA) by the end of the experiment despite continuous antibiotic treatment. This is consistent with the current knowledge that following an initial reduction in bacterial load, antibiotic-resilient bacterial communities will repopulate the gut community [20].

The primary goal of this study was to determine the extent to which gut microbe-altering antibiotic treatment alleviated fatigue-like behaviors (reduced locomotion and activity) that are observed in paclitaxel-treated mice, thereby implicating the gut microbiota in chemotherapy-induced fatigue. Consistent with our previous data [4, 15] paclitaxel chemotherapy reduced running wheel activity and locomotion in an open field chamber. This is also consistent with the data of others demonstrating a reduction in dark phase horizontal locomotor activity and wheel running [21]. However, some groups have not observed a reduction in locomotion in mice treated with paclitaxel [22, 23], likely due to differences between our higher-

dose paclitaxel (cumulative 180 mg/kg) and those with a lower dose of paclitaxel (cumulative 4 or 80, mg/kg, respectively). Centrally-mediated fatigue, as experienced with CRF, is a multidimensional condition that encompasses side effects including decreased cognitive function and motivation [3]. Measurement of various dimensions of central fatigue in rodent models can include measurements of activity, exploration, reward seeking, and motivation [3]. Two measurements of fatigue-like behavior were used in this study, voluntary exercise (i.e., running wheels) and movement in an open field. The inclusion of these two measures allows us to demonstrate that paclitaxel affects both passive physical activity (decreased activity in an open field) and motivated behaviors (decreased voluntary wheel running [24]). Additional aspects of fatigue, including reward motivation/effort, could be assessed in future studies using operant behavior tasks. Others have shown that paclitaxel treatment decreases sucrose preference in mice [22], indicating a potential multidimensional phenotype of fatigue-like behavior. The ability of antibiotic treatment to ameliorate fatigue-like behavior in an open field implicates the gut microbiome in the development of paclitaxel-induced reduced activity. Observed in both behavioral tasks, is a reduction in activity in Abx-Veh mice compared to Control-Veh mice. This would suggest that disruption of a healthy gut microbiome is detrimental and may cause fatigue behaviors. It is also notable that, by the completion of this experiment, all animals had roughly the same running wheel activity thereby suggesting that paclitaxel-associated fatigue recovers over time.

Lastly, we examined the potential interaction of paclitaxel and antibiotic treatments on energy homeostasis and melanocortin signaling as a potential mechanism leading to the ability of antibiotics to attenuate paclitaxel-associated activity deficits. We first investigated this signaling pathway in the context of paclitaxel chemotherapy and reported that exercise promoted the recovery of chemotherapy-associated weight loss, increased circulating ghrelin and decreased circulating leptin, and restored hypothalamic *Pomc* expression in paclitaxel-treated mice [4]. This implication of the melanocortin signaling pathway in the development of chemotherapy-associated sickness behaviors, led to our current study on fatigue. Following paclitaxel recovery, antibiotic treatment increased plasma ghrelin. This alteration in plasma ghrelin was anticipated, as the gut microbiome and microbial metabolites are known to affect ghrelin concentrations and signaling (reviewed in Leeuwendaal, et al., 2021 [25]). It should be noted that this is a measure of total ghrelin and not acylated ghrelin, the active form. Additionally, the most accurate measurements of leptin and ghrelin are taken in plasma from mice that have been food restricted. While these mice were provided food and water ad libitum, blood samples were obtained 1 day post paclitaxel treatment at 1300, during the light phase of their light/dark cycle when feeding should be minimal. Eight days post paclitaxel treatment samples were taken in the early dark phase. After paclitaxel recovery, while mice were still eating antibiotic chow, we observed an increase in ghrelin and decrease in leptin caused by antibiotics. These changes are likely due to gut microbiota changes as a study of male rats shows a negative correlation between *Bacteroides* and leptin serum levels across multiple nutritional statuses [26] and we have previously observed a decrease in *Bacteroides* with this antibiotic treatment [15]. Additional changes to the melanocortin system were induced by antibiotic treatment in the form of decreasing the expression of *Pomc* and *Htr2c* in the hypothalamus. This, in combination with the reduction in leptin, is consistent with reduced satiety signals. This is reflected by the increase in food intake of antibiotic-fed mice that occurs between day 12 and day 20.

The results of this study are modest and therefore should be regarded in the context of other bodies of work. Future studies will need to address these limitations to significantly impact the understanding of how the gut microbiome plays a role in the development of chemotherapy-associated fatigue. This study was conducted only in female mice and with relatively small sample sizes (n = 6-8/group), therefore the statistical power was limited and

variability of the biology and activity results were significant. While paclitaxel chemotherapy is used in the treatment of cancer, mice in this study were tumor-naive. Including tumor-bearing mice would provide a more complete understanding of fatigue in cancer patients. The antibiotic treatment used in this study targeted both gram-positive and -negative bacteria and was administered for the entirety of the study to modify the gut microbiome. Future manipulation of the antibiotics regimen and schedule could address the potential for antibiotics, used periodically to resolve infection in chemotherapy patients, to modulate chemotherapy-associated fatigue and energy homeostasis. Furthermore, bacterial sequencing was not repeated for these mice as in our previous study which demonstrated both antibiotic and chemotherapy consequences for the gut relative bacterial taxa abundances [15]. Future bacterial sequencing would allow for correlation analyses between locomotion, hypothalamic PCR, hormone plasma concentrations, and specific gut taxa. Lastly, chemotherapy-associated side effects are known to be persistent in some patients, therefore the focus of this study acutely after treatment does not capture the potential effects of antibiotics on long-term side effects of paclitaxel.

## Conclusions

Here we have demonstrated a modest modulation of chemotherapy-associated fatigue-like behaviors in mice by the gut microbiota. Moderate alterations of energy homeostatic hormones (i.e., leptin and ghrelin) are supported as one mechanism for this gut-directed, antibiotic-mediated modulation. Given the modest sample size (n = 6–8), this work may be regarded as preliminary data for future studies. Replication of these findings, further investigation of the effects of antibiotics on reward motivation and effort aspects of fatigue, and studying fatigue caused by other types of commonly used chemotherapeutics will all be important questions for future work. This current study builds upon existing literature that implicates the gut microbiota in the development of chemotherapy-associated behavioral side effects, thus encouraging the development of gut-directed interventions for the treatment of these side effects.

## Materials and methods

### Mice

Female, 12-week old, nulliparous, C57BL/6 mice (Charles River, Wilmington, MA, USA) were initially housed 5/cage, handled every other day, and acclimated for at least 1 month to a 14:10 light:dark cycle (lights off at 1400 h) in a temperature-controlled vivarium (22 ± 1˚C) at The Ohio State University. Mice were singly-housed when in-cage running wheels were added (see *Voluntary wheel running*). Chow and water were available ad libitum throughout the duration of the study unless otherwise indicated. To measure food intake, food weight was taken and averaged for a g/24h unit of measurement.

### Ethics statement

All animal experiments were carried out in strict accordance with the National Institutes of Health Guide for the Care and Use of Laboratory Animals [27]. All efforts were made to minimize animal suffering and to reduce the number of mice used. The protocol was approved by the Ohio State University's Institutional Animal Care and Use Committee (Protocol Number: 2017A00000093-R2).

## Experimental design overview

This study was completed in 2 waves of 3–4 mice per group. Treatment groups consisted of Control-Vehicle, Antibiotics-Vehicle (Abx-Vehicle), Control-Chemotherapy (Control-Chemo), and Antibiotics-Chemotherapy (Abx-Chemo). All mice were given running wheels and experimental chow (control or antibiotics) for the duration of the study as outlined in **Fig 1A**. The day following the last vehicle or chemotherapy administration, retro-orbital blood was collected. Two days following the last vehicle or chemotherapy administration, the open field test was conducted. Mice recovered from treatment for an additional 6 days while remaining on experimental chow, followed by tissue collection.

## Antibiotics

An antibiotic cocktail consisting of 0.5 g/kg vancomycin, 1 g/kg metronidazole, and 1 g/kg neomycin trisulfate was administered via integration into non-irradiated Teklad LM-485 chow (Envigo, Madison, WI, USA). This treatment was previously demonstrated to significantly decrease gut bacterial alpha diversity and reduce the relative abundance of about 100 genera in the colon [15]. Antibiotic or control chow was provided for 10–11 d prior to the beginning of chemotherapy treatment and throughout the duration of the study.

## Chemotherapy

Paclitaxel (>97%, Sigma-Aldrich, St. Louis, MO, USA) was dissolved in a 50:50 mixture of Cremophor-EL and 200-proof EtOH, then diluted 1:1 with sterile PBS. Mice were injected with paclitaxel or vehicle using the previously-reported, clinically-relevant injection protocol of 30 mg/kg, I.P. every other day for a total of 6 doses [4, 15, 28]. Body mass and food intake were measured at each injection and during recovery to assess cachexia and anorexia. Per IACUC guidelines, animals were immediately euthanized if they lost >20% of their baseline body mass (n = 2).

## Voluntary wheel running

In-cage running wheels (Starr Life Sciences, Oakmont, PA, USA) with a 120.7 mm diameter and 50.8 mm width were placed in mouse cages for at least 2 weeks prior to the start of chemotherapy treatment. Glass probes recorded the number of wheel rotations binned at 15-sec sampling intervals using VitalView 5.1 software (Starr Life Sciences). Dark phase (1400–2399) revolutions were aggregated. Data are presented as a percent of baseline dark phase revolutions (baseline was the average of 3 d prior to starting chemotherapy).

## Open field

In the early part of the dark phase (1430–1800 h), mice were placed in one corner of a 16" x 16" photobeam arena (San Diego Instruments, San Diego, CA, USA) lightly coated with corn-cob bedding, and allowed to explore for 15 min to measure general locomotion in a novel open field. The apparatus was cleaned with 70% ethanol between mice. Locomotion was analyzed using PAS Data Reporter (San Diego Instruments) and reported as ambulatory beam breaks.

## Tissue collection

One day after the treatment regimen was completed, mice were anesthetized with vaporized isoflurane at 1300 h (late light phase) and a small, heparin-lined micro-capillary tube was used to collect <200 μL of blood from the retro-orbital sinus. Whole blood was stored on ice until

plasma could be collected by centrifuging for 20 min at 900 RCF at 4 ˚C. One week later, tissue collections occurred in the early dark phase (1400–1800 h). Mice were euthanized by rapid decapitation and trunk blood collected via a heparin-lined Natelson tube and plasma was extracted. The hypothalamus was fresh dissected and frozen on dry ice. All tissues were stored at -80 ˚C until the time of analysis.

## Plasma hormone concentrations

Total ghrelin and leptin were measured in plasma using a custom U-PLEX immunoassay (Meso Scale Discovery, Rockville, MD, USA) according to the manufacturer's instructions. Plasma samples were thawed on ice and ran in duplicate, when possible, on a QuickPlex SQ 120 instrument (Meso Scale Discovery). Nineteen of 25 samples from one day post drug treatment and 16/30 samples from final tissue collection were run in singlicate because only one well worth of plasma (50 μL) was available. Values determined to be "not detected" were reported as zero (Control-Vehicle, n = 1 on both days). The intra-assay variability was ≤10% for leptin and ≤20% for ghrelin.

## Quantitative RT-PCR

Total RNA was extracted from the hypothalamus using Qiagen RNeasy Mini kits (Qiagen, Germantown, MD, USA) and reverse transcribed using qScript cDNA Supermix (Quantabio, Beverly, MA, USA). Gene expression was assessed using TaqMan Fast Advanced Master Mix (Thermo Fisher Scientific) and TaqMan probes (*Gapdh*: Mm99999915_g1, *Hprt*: Mm03024075_m1, *Ghsr*: Mm00616415_m1, *Hcrtr1*: Mm01185776_m1, *Npy*: Mm01410146_m1, *Pomc*: Mm00435874_m1, *Htr2c*: Mm00434127_m1). Results are expressed at $2^{-\Delta Ct}$ normalized to the geometric mean of the Ct value of *Gapdh* and *Hprt*. The geometric mean of *Gapdh* and *Hprt* were not significantly different between treatment groups.

## Statistical analyses

A repeated measures mixed-effect model was used to determine the main effects in body mass, food intake, wheel revolution, and total locomotion data. Missing values from wheel revolutions (1 day of probe malfunction) were imputed using a simple unconditional mean imputation method [29]. A two-way ANOVA with Tukey's multiple comparison post-hoc test was used to determine significant effects in average revolution, leptin, and ghrelin. A two-way ANOVA with post-hoc t-test was used to determine significant effects in open field total locomotion and hypothalamic gene expression data. Outliers were removed via Grubb's method with $\alpha = 0.05$. P values are reported for main effects, for multiple comparisons, *$p \leq 0.05$, **$p \leq 0.01$, ***$p \leq 0.001$, ****$p < 0.0001$.

## Supporting information

**S1 Fig. Effect of antibiotic treatment on 16s rRNA.**
(DOCX)

## Acknowledgments

The authors thank Selina Vickery, Browning Haynes, and Emma Siefring for their technical assistance. The authors also thank Megan Fleming, Cindy Fairbanks, and Dr. Dondre Coble for animal husbandry.

## Author Contributions

**Conceptualization:** Kelley Jordan, Leah M. Pyter.

**Data curation:** Kelley Jordan, Melina M. Seng.

**Formal analysis:** Corena V. Grant, Melina M. Seng.

**Funding acquisition:** Leah M. Pyter.

**Visualization:** Melina M. Seng.

**Writing – original draft:** Corena V. Grant.

**Writing – review & editing:** Corena V. Grant, Kelley Jordan, Melina M. Seng, Leah M. Pyter.

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
