## [Decision Letter · Decision Letter 0]

19 Jul 2022

PONE-D-22-14575Antibiotic treatment inhibits paclitaxel chemotherapy-induced activity deficits in female mice

PLOS ONE

Dear Dr. Pyter,

Thank you for submitting your manuscript to PLOS ONE. After careful consideration, we feel that it has merit but does not fully meet PLOS ONE’s publication criteria as it currently stands. Therefore, we invite you to submit a revised version of the manuscript that addresses the points raised during the review process.

We look forward to receiving your revised manuscript.

Kind regards,

Neelu Jain Gupta, Ph.D.

Academic Editor

PLOS ONE

Journal Requirements:

2. Please expand the acronym “NIH ” (as indicated in your financial disclosure) so that it states the name of your funders in full.

   "This work was supported by NIH grant CA216290 and an associated personnel supplement -04S2 (L.P., C.G. [trainee])"

  "The authors thank Selina Vickery, Browning Haynes, and Emma Siefring for their technical assistance. The authors also thank Megan Fleming, Cindy Fairbanks, and Dr. Dondre Coble for animal husbandry. This work was supported by NIH grant CA216290 and an associated personnel supplement -04S2 (L.P., C.G. [trainee])"

 "This work was supported by NIH grant CA216290 and an associated personnel supplement -04S2 (L.P., C.G. [trainee])"

Reviewers' comments:

Reviewer's Responses to Questions

**Comments to the Author**

1. Is the manuscript technically sound, and do the data support the conclusions?

Reviewer #1: Yes

Reviewer #2: No

2. Has the statistical analysis been performed appropriately and rigorously? 

Reviewer #1: Yes

Reviewer #2: No

3. Have the authors made all data underlying the findings in their manuscript fully available?

Reviewer #1: Yes

Reviewer #2: Yes

4. Is the manuscript presented in an intelligible fashion and written in standard English?

Reviewer #1: Yes

Reviewer #2: Yes

5. Review Comments to the Author

Reviewer #1: The manuscript by Grant et al. employs antibiotics to modulate chemotherapy-induced behavioral deficits in mice, providing additional evidence that the gut microbiota likely contributes to the development of chemotherapy-associated side effects as they demonstrated in previous studies. In the current study, the authors focused on locomotor activity, weight loss/feeding behavior and endocrine responses. The study is straightforward and extends their previous work.

Major

1. Fig 1. Was body weight during the chow/wheel running prior to chemo not monitored? This information would help support the current findings from chemo day 1 until the end of the experiment to see if weight was stable or impacted by abx prior to treatment.

2. Fig 1. How was food intake measured? This is typically done in a metabolic cage, which is known to be stressful. In this case, stress would be an additional variable to consider in the context of weight loss and behavior.

3. Figure 2. It is unclear why the authors highlight the dark phase revolutions on the day of the open field. What is the significance of this data to the overall effect? How is the significant decrease overall in activity caused by the OFT impacting the differences highlighted between the treatment groups? For example, it appears that a decrease in revolutions is caused by ABX in the vehicle group on open field day that was not observed in the average of the treatment days – what does this reduction in activity reflect in the abx group? Perhaps this simply reflects the choice of study design.

The left panel of 2A is hard to read as currently presented and it would be recommended to make the x-axis longer to allow visualization of the points more clearly. Since the ABX+vehicle appear to have a decrease in activity on the chemo days despite not having chemo it would appear that some of the effect is investigator driven (initiated by the presence of the investigator treating other animals). This should be discussed.

Panel C – units would be helpful to understand the panel easier.

Why didn’t the authors include additional OFT parameters such as total distance traveled, or time spend in the periphery/center of arena to gain additional insight into anxiety-like behavior or total movement?

4. Why were leptin/ghrelin samples taken at different points in the L/D cycle? If animals were not fasted as is typically performed in these measurements, why were day 8 samples not taken during the light phase? Would it not be interesting to compare the impacts of these two timepoints within each group, rather than the effects across groups? This analysis was not included.

Minor

1. What is ClpB? Not currently defined.

2. Please avoid use of “tended” to describe changes. If they are not significant, they are not different. If they are “trending” than be sure to highlight the rationale for pointing out a non-significant effect, including low n number or high variability.

3. “Chemotherapy and ABX energy-regulating systems” needs to be clarified. Modulate perhaps is what the authors meant?

Reviewer #2: 1. Author claims that antibiotic cocktail treatment alters the gut microbiota in chemotherapy treated mice which in turn attenuates the fatigue like behavior by modulating the melanocortin axis. The major concern is that author has not shown any microbial consortium in these mice model. Authors should show the microbial diversity in every conditions.

2. Chemotherapy and antibiotics (Abx-chemo) both treatments cause the decline in body weight. Is there any significant difference in between these two treatments? In both the cases it starts to recover after 16 and become almost similar on later time points. What is the most probable region for this enhancement?

3. In methodology section, author mentioned the reatment groups consisted of Control-Vehicle, Antibiotics-Vehicle (Abx-Vehicle), Control-Chemotherapy (Control-Chemo), and Antibiotics-Vehicle (Abx-Chemo). What is the difference in between the Antibiotics-Vehicle (Abx-Vehicle) and Antibiotics-Vehicle (Abx-Chemo). If both are similar why two different symbols have been used.

4. In fig.2, wheel running activityalso showing similarity in control-chemo and Abx-chemo group. It begins to increase and was almost similar in all groups on day 19. Why such changes have been happened.

5. Is there any mortality in this experiment?

6. It will be good if author would measure the level of POMC protein level in serum.

6. PLOS authors have the option to publish the peer review history of their article (what does this mean?). If published, this will include your full peer review and any attached files.

Reviewer #1: No

Reviewer #2: **Yes: **Nitin Bhardwaj

---

## [Author Response · Author response to Decision Letter 0]

5 Aug 2022

The authors would like to thank the reviewers for taking the time to critically review and provide comments on this manuscript. Each comment has been considered and revisions made to the manuscript accordingly. The authors believe that, because of these comments, the manuscript has been significantly improved. Below, please find the responses to specific reviewer comments. 

Reviewers' comments:

Reviewer #1

The manuscript by Grant et al. employs antibiotics to modulate chemotherapy-induced behavioral deficits in mice, providing additional evidence that the gut microbiota likely contributes to the development of chemotherapy-associated side effects as they demonstrated in previous studies. In the current study, the authors focused on locomotor activity, weight loss/feeding behavior and endocrine responses. The study is straightforward and extends their previous work.

1. Fig 1. Was body weight during the chow/wheel running prior to chemo not monitored? This information would help support the current findings from chemo day 1 until the end of the experiment to see if weight was stable or impacted by abx prior to treatment.

The authors agree that body weight and food intake information prior to the beginning of vehicle or chemotherapy treatment would be useful for interpretation. Figure 1 has been updated as such.

The results and discussion have also been adjusted accordingly. Briefly, lines 77-90 in the results detail that during baseline, antibiotic chow consumption was initially lower than control chow; however, by the time vehicle or control treatment began, both chow groups were eating the same amount. This pattern was likewise observed for baseline body mass. However, during treatment and recovery, antibiotic chow consumption again fell below that of control chow, although body mass was only affected by chemotherapy during this period. 

2. Fig 1. How was food intake measured? This is typically done in a metabolic cage, which is known to be stressful. In this case, stress would be an additional variable to consider in the context of weight loss and behavior.

For this experiment, mice were singly housed in standard cages and provided with a running wheel that acts as both a measure of activity and a form of enrichment. This allowed us to simply weigh the food and calculate how much a single animal had consumed over the time period, then average that for a g/24 h unit of measurement. Care was taken to include food pieces that were scattered on the floor of the cage in the food weight. This has been included on line 225 in the “Materials and Methods” section.

3. Figure 2. It is unclear why the authors highlight the dark phase revolutions on the day of the open field. 

a. What is the significance of this data to the overall effect? 

We have highlighted in the text on lines 98 and 107 that wheel running was assessed during the dark phase because that is when mice are most active (line 98). We chose a focused visualization of wheel behavior during the dark phase on the same day/time that corresponds with the open field data (line 107). This is important because it shows congruence between fatigue behaviors as measured by running wheels and open field. 

b. How is the significant decrease overall in activity caused by the OFT impacting the differences highlighted between the treatment groups? For example, it appears that a decrease in revolutions is caused by ABX in the vehicle group on open field day that was not observed in the average of the treatment days – what does this reduction in activity reflect in the abx group? Perhaps this simply reflects the choice of study design.

The left panel of 2A is hard to read as currently presented and it would be recommended to make the x-axis longer to allow visualization of the points more clearly. Since the ABX+vehicle appear to have a decrease in activity on the chemo days despite not having chemo it would appear that some of the effect is investigator driven (initiated by the presence of the investigator treating other animals). This should be discussed.

Panel C – units would be helpful to understand the panel easier.

Why didn’t the authors include additional OFT parameters such as total distance traveled, or time spend in the periphery/center of arena to gain additional insight into anxiety-like behavior or total movement?

As the reviewer points out, removing mice from the wheel running cage to complete the OFT does cause a slight reduction in overall activity, however, mice are returned immediately to wheel cages after the OFT to minimize the time away. Additionally, the effect of completing OFT will have affected all groups equally. 

The size of Figure 2A, left panel has been increased to improve visibility.

The authors agree, that there is an effect of receiving injections on all groups, demonstrated by the reduction in wheel revolution compared to the recovery days prior. However, the more pronounced reduction in Abx-Veh vs Control-Veh is attributed to also receiving antibiotics, which can cause some minor sickness behaviors (see food intake). This is now included in the Discussion on line 180: “Observed in both behavioral tasks, is a reduction in activity in Abx-Veh mice compared to Control-Veh mice. This would suggest that disruption of a healthy gut microbiome is detrimental and may cause fatigue behaviors.”

 The y-axis has been adjusted to include units in Fig 2C.

The scope of this short communication focused on fatigue behaviors. We have explored anxiety-like behavior in a previous publication (Grant et al., 2021, Brain, Behav., Immun.). When using automatic photobeam chambers such as the ones used in this study as compared to a rodent tracking system, beam breaks are more accurate than measures of distance.

4. Why were leptin/ghrelin samples taken at different points in the L/D cycle? 

a. If animals were not fasted as is typically performed in these measurements, why were day 8 samples not taken during the light phase? Would it not be interesting to compare the impacts of these two timepoints within each group, rather than the effects across groups? This analysis was not included.

The reviewer makes a good point that the tissues are not collected at the same time point and therefore comparisons cannot be made across time points. However, the decision to collect blood from mice 1 day post chemotherapy in the late light phase (1300) was made to minimize the impact of RO eye bleed on dark phase animal behavior. However, the final tissue collection (8 days post chemotherapy) occurred in the early dark phase (1400-1800) to capture biological samples that reflect the most active point in animal behavior. Given the circadian rhythmicity of these two hormones (Kalra, et al., 2003, Regulatory Peptides), the interpretation of a comparison of these two timepoints within group would be hazardous.

5. What is ClpB? Not currently defined.

The full name for ClpB (caseinolytic peptidase B protein homolog) is now included on line 58 and the abbreviation has been eliminated because this protein is only mentioned once in the manuscript.

6. Please avoid use of “tended” to describe changes. If they are not significant, they are not different. If they are “trending” then be sure to highlight the rationale for pointing out a non-significant effect, including low n number or high variability.

Tended was used in 2 instances in this manuscript and it has been removed in both locations. First, it was used to describe the data that led to a Chow x Treatment effect in wheel running when there were no significant comparisons using a Dunnett’s Multiple Comparisons test completed after the 2-way repeated measures ANOVA. This portion on line 101 now reads “Antibiotics interacted with chemotherapy treatment (Fig 2A, Chow x Treatment effect, F1,25=5.4, p<0.05), driven by decreased activity in antibiotic versus control chow-fed mice in within the vehicle treatment group, but increased activity in antibiotic versus control chow-fed mice in within the chemotherapy treatment group.”

The second location is in reference to a statistical analysis resulting in a p=0.06 value, and it has been adjusted in line 124 as follows: “One day after treatment completion, a trend for chemotherapy to increase ghrelin was observed; this likely did not reach significance because of high variability in the Control-Chemo group (Fig 3A, Treatment effect F1,21=4.0, p=0.06).”

7. “Chemotherapy and ABX energy-regulating systems” needs to be clarified. Modulate perhaps is what the authors meant?

We agree this needed some clarification, the title of this section has been adjusted on line 122 to “Chemotherapy and antibiotics modulate energy-regulating systems”

Reviewer #2:

8. Author claims that antibiotic cocktail treatment alters the gut microbiota in chemotherapy treated mice which in turn attenuates the fatigue like behavior by modulating the melanocortin axis. The major concern is that author has not shown any microbial consortium in these mice model. Authors should show the microbial diversity in every conditions.

While microbial diversity is not reported in these specific animals, this exact antibiotic treatment regimen was used in a previous publication of ours (Grant et al., 2021). That publication reports an overall reduction in alpha diversity and reductions in the relative abundance of approximately 100 genera. This is included in the Methods section of the manuscript on line 243.

9. Chemotherapy and antibiotics (Abx-chemo) both treatments cause the decline in body weight. Is there any significant difference in between these two treatments? In both the cases it starts to recover after 16 and become almost similar on later time points. What is the most probable reason for this enhancement?

During the treatment period, there is only a decline in body mass associated with chemotherapy treatment, which occurs regardless of chow treatment. The body mass recovery in the Control-Chemo and Abx-Chemo groups is likely a result of cessation of chemotherapy treatment. To address this comment and comment 1 by Reviewer 1, the results section on body mass has been expanded on lines 77-90.

10. In methodology section, author mentioned the treatment groups consisted of Control-Vehicle, Antibiotics-Vehicle (Abx-Vehicle), Control-Chemotherapy (Control-Chemo), and Antibiotics-Vehicle (Abx-Chemo). What is the difference in between the Antibiotics-Vehicle (Abx-Vehicle) and Antibiotics-Vehicle (Abx-Chemo). If both are similar why two different symbols have been used.

We appreciate the reviewer catching this mistake in nomenclature. This is a 2x2 study design and “Vehicle” has been changed to “Chemotherapy” as such on line 233.

11. In fig.2, wheel running activity also showing similarity in control-chemo and Abx-chemo group. It begins to increase and was almost similar in all groups on day 19. Why such changes have been happened.

Yes, there is definitely a similarity between Control-Chemo and Abx-Chemo groups because of the strong fatigue behavior induced by chemotherapy treatment. However, as highlighted in Fig 2A middle and right panel, there is only a significant difference between vehicle and chemotherapy treatment in the control chow group, not the antibiotic chow group, demonstrating that antibiotics modulates this chemotherapy effect. The similar wheel running on day 19 is likely driven by a recovery from treatment (chemotherapy treated groups) and the absence of stress associated with injections (vehicle group). This has been added to the Discussion on line 183 with the sentence “It is also notable that, by the completion of this experiment, all animals had roughly the same running wheel activity thereby suggesting that chemotherapy-associated fatigue recovers over time.”

12. Is there any mortality in this experiment?

Two animals were euthanized because they lost greater than 20% of their body mass with chemotherapy treatment. This is stated in the Methods on line 252.

13. It will be good if author would measure the level of POMC protein level in serum.

This is a great suggestion, unfortunately, after running the Leptin/Ghrelin assay, there is not enough blood left to complete this.

---

## [Decision Letter · Decision Letter 1]

27 Sep 2022

PONE-D-22-14575R1Antibiotic treatment inhibits paclitaxel chemotherapy-induced activity deficits in female micePLOS ONE

Dear Dr. Pyter,

Thank you for submitting your manuscript to PLOS ONE. After careful consideration, we feel that it has merit but does not fully meet PLOS ONE’s publication criteria as it currently stands. Therefore, we invite you to submit a revised version of the manuscript that addresses the points raised during the review process.

 One of the reviewers is satisfied with the revision, however, the other reviewer who was brought in has noted a number of significant issues that need to be addressed in a revised manuscript. Importantly, there is concern that the differences are modest at best and the statistical analysis of the results is not clearly presented to support the significance of the results.

We look forward to receiving your revised manuscript.

Kind regards,

Brenda A Wilson, Ph.D.

Academic Editor

PLOS ONE

Additional Editor Comments (if provided):

This manuscript has an interesting premise, and for the most part the experiments appear to be done appropriately. But the results are unremarkable, and the differences are modest. Although the authors claim to have support for their hypothesis, the statistical analysis of the results is not clearly presented in a way that is understandable.

Reviewers' comments:

Reviewer's Responses to Questions

**Comments to the Author**

1. If the authors have adequately addressed your comments raised in a previous round of review and you feel that this manuscript is now acceptable for publication, you may indicate that here to bypass the “Comments to the Author” section, enter your conflict of interest statement in the “Confidential to Editor” section, and submit your "Accept" recommendation.

Reviewer #1: All comments have been addressed

Reviewer #3: (No Response)

2. Is the manuscript technically sound, and do the data support the conclusions?

Reviewer #1: Yes

Reviewer #3: Partly

3. Has the statistical analysis been performed appropriately and rigorously? 

Reviewer #1: Yes

Reviewer #3: I Don't Know

4. Have the authors made all data underlying the findings in their manuscript fully available?

Reviewer #1: Yes

Reviewer #3: Yes

5. Is the manuscript presented in an intelligible fashion and written in standard English?

Reviewer #1: Yes

Reviewer #3: Yes

6. Review Comments to the Author

Reviewer #1: Thank you for thoroughly responding to all reviewer comments.

Reviewer #3: The authors treated mice with a chemotherapy regimen based upon paclitaxel in combination with antibiotics to test whether gut microbes contribute to chemotherapy-associated fatigue. Only female mice were included in the study. The authors state that the data presented support their claim. However, only a subset of the parameters showed differences and these effects were very modest. This study would be improved by increasing the number of mice included in each group because most of the results have large variances, by examining the effects of antibiotic treatment for a longer time after termination of chemotherapy and after terminating antibiotic treatment at the same time as termination of chemotherapy.

Throughout the manuscript the authors need to clearly state which chemotherapy regimen was used in the study. Presently, the exact chemotherapy is only stated in the title of the paper and infrequently in the methods. It is well established that different types of chemotherapy induce specific side effects. The potential effects on different chemotherapy regimens must be discussed.

Regardless of prior reports, verification that the gut microbiome was altered by the antibiotic treatment is needed. Were any bacteria remaining? One antibiotic treatment was terminated, how fast did bacteria return and which bacteria returned? Is there a difference if the antibiotic treatment was terminated at the same time as the chemotherapy?

The decreased food intake with antibiotic treatment is a confounding factor in these studies. More controls for food intake are needed. In figure 1 the body mass decreased with chemotherapy with and without antibiotics, as would be expected. But the food intake (panel C) does not match this difference. During treatment, both antibiotic treated groups have decreased food intake. More discussion about this discrepancy is needed.

Figure 2A shows no effect of the antibiotic treatment on the number of revolutions (compare values shown as open and closed red curves). And the right bar graph of panel 2A shows no improvement with the addition of antibiotic treatment in the paclitaxel treated mice. Protection with antibiotic treatment may be present in the open field test, but the number of mice included is small and the effect is quite modest. Is the difference between the two groups - chemotherapy with and without antibiotics – significantly different? Regardless, more mice need to be tested before a conclusion can be asserted because the variance is very large.

In figure 3, the main effects on ghrelin and leptin (panels A and B) appears to be changes with antibiotic treatment, regardless of the presence of paclitaxel treatment. At one day post treatment there may be modest differences, but at 8 days post treatment, no differences are presented. Panels C-G also only show antibiotic effects, but no difference with and without chemotherapy.

Need to compare the results presented here with previously published results showing that paclitaxel treatment did not alter locomotion (for example, Nguyen et al, Molecular Neurodegeneration, 2021).

The present study only examined the effects that persisted 8 days after completion of treatment. Many side effects of chemotherapy persist for prolonged periods. The authors need to include experiments that address the long duration of the side effects. Does the proposed protection persist after terminating chemotherapy only when antibiotic treatment is maintained or does protection persist when both chemotherapy and antibiotic treatment terminate at the same time?

Does the proposed protection alter the ability to treat tumor size and/or spread to additional tissues?

In the statistical measurements shown on the figures, it is not clear which bars are being compared. Figures should be easily understood, without the need to read the entire text. Please include clarification on the figures.

7. PLOS authors have the option to publish the peer review history of their article (what does this mean?). If published, this will include your full peer review and any attached files.

Reviewer #1: No

Reviewer #3: No

---

## [Author Response · Author response to Decision Letter 1]

2 Dec 2022

A "Response to Reviewers" document has been attached with the same information provided here plus a graph which cannot be included here. 

In addition to the original revisions that we previously submitted in response to Reviewers 1 and 2, we appreciate the time and effort Reviewer #3 has taken to review our manuscript. Please see our detailed responses below for how comments were incorporated into the manuscript. Reviewers 1 and 2 comments were deemed as adequately addressed in the first round of revisions.

1. Throughout the manuscript the authors need to clearly state which chemotherapy regimen was used in the study. Presently, the exact chemotherapy is only stated in the title of the paper and infrequently in the methods. It is well established that different types of chemotherapy induce specific side effects. The potential effects on different chemotherapy regimens must be discussed.

Throughout the manuscript, “chemotherapy” was replaced with “paclitaxel,” the specific chemotherapeutic agent used in this study. 

In the Conclusions, the following is now included beginning on line 226.

“Given the modest sample size, future studies should replicate these findings, further investigate the effect of antibiotics on reward motivation and effort aspects of fatigue, and study fatigue caused by other types of commonly used chemotherapeutics.”

2. Regardless of prior reports, verification that the gut microbiome was altered by the antibiotic treatment is needed. Were any bacteria remaining? One antibiotic treatment was terminated, how fast did bacteria return and which bacteria returned? Is there a difference if the antibiotic treatment was terminated at the same time as the chemotherapy?

We have previously published 16S sequencing of bacterial community DNA using this antibiotic and chemotherapy paradigm (Grant et al., 2021). In the Methods, we added a statement with the reference to this previous comprehensive gut bacterial analysis as follows: “This treatment was previously demonstrated to significantly decrease gut bacterial alpha diversity and reduce the relative abundance of about 100 genera in the colon.[15]”

Fecal samples from the present study were not collected from all mice at all points of the experiment and budget restrictions prevented re-sequencing, however, we were able to complete quantification of the total fecal bacterial 16S gene from a subset of these mice. Consistent with our previous work, the graph here demonstrates that mice that received antibiotic chow had a lower amount of gut bacteria then mice that received control chow (*p=0.03) and compared to pre-chow gut bacteria content (& p=0.01). The recovery of overall bacterial levels at the end of the experiment is likely the result of antibiotic-resistant bacterial taxa blooming in the gut as this analysis represents total bacteria, irrespective of relative abundances of specific taxa. Indeed antibiotics reduce diversity of the gut microbial community without wiping out all types of bacteria; some thrive when others are depleted (Lange et al., 2016). 

3. The decreased food intake with antibiotic treatment is a confounding factor in these studies. More controls for food intake are needed. In Figure 1 the body mass decreased with chemotherapy with and without antibiotics, as would be expected. But the food intake (panel C) does not match this difference. During treatment, both antibiotic treated groups have decreased food intake. More discussion about this discrepancy is needed.

This study was designed as a standard 2 x 2 experimental design with a total of 4 groups +/- Antibiotic chow and +/- Paclitaxel chemotherapy, therefore we believe this study is properly controlled. The authors agree that the dichotomy between food intake and body mass is puzzling, however, chemotherapy and antibiotic treatment affects metabolism thereby changing the expectation that an increase in food intake leads to an increase in weight. 

This is thoroughly discussed for readers beginning on line 154 in the Discussion. [The underlined section was added to the original discussion].

“As previously published, this paclitaxel chemotherapy regimen transiently reduced body mass compared to vehicle-treated mice,[18] which was not due to a difference in food intake. However, antibiotic-treated mice ate significantly less chow than control mice without an impact on body mass. It is probable that this is due to the aversive taste of antibiotics because the reduction is observed throughout the entire experiment, and others have shown that mice reduce water intake when the water is formulated with a similar antibiotic cocktail containing ampicillin, vancomycin, neomycin, and metronidazole.[19] Interestingly, during baseline acclimation of food and running wheels, mice initially consumed less antibiotic-containing food, however by the time of vehicle or paclitaxel treatment, that was not the case. During the course of treatment, though, mice again consumed less antibiotic-containing chow, however weight loss was only impacted by chemotherapy treatment, not a decrease in antibiotic food intake. This is likely due to the metabolic changes documented in patients treated with chemotherapy.[20] Further, the increase in ghrelin and decrease in leptin caused by antibiotics (measured 8 days post treatment) would suggest food intake decreases are occurring despite appetite-stimulating signals. Despite the reduced food intake, we have previously demonstrated that this antibiotic chow is effective at decreasing gut bacterial alpha diversity.[15]”

4. Figure 2A shows no effect of the antibiotic treatment on the number of revolutions (compare values shown as open and closed red curves). And the right bar graph of panel 2A shows no improvement with the addition of antibiotic treatment in the paclitaxel treated mice. Protection with antibiotic treatment may be present in the open field test, but the number of mice included is small and the effect is quite modest. Is the difference between the two groups - chemotherapy with and without antibiotics – significantly different? Regardless, more mice need to be tested before a conclusion can be asserted because the variance is very large.

As indicated in Figure 2, there is not a significant difference between Control-Chemo and Abx-Chemo in either wheel running or open field. However, this specific post-hoc analysis type of question is not the main goal of the study. The goal of this study was to identify the main effects and interactions of Antibiotics and Chemotherapy treatment. Unfortunately, the addition of mice to the study at this time is beyond our budget. Replication of these findings in future studies is now listed in the Conclusions beginning on line 226. 

“Given the modest sample size, future studies should replicate these findings, further investigate the effect of antibiotics on reward motivation and effort aspects of fatigue, and study fatigue caused by other types of commonly used chemotherapeutics.”

5. In Figure 3, the main effects on ghrelin and leptin (panels A and B) appears to be changes with antibiotic treatment, regardless of the presence of paclitaxel treatment. At one day post treatment there may be modest differences, but at 8 days post treatment, no differences are presented. Panels C-G also only show antibiotic effects, but no difference with and without chemotherapy.

This is correct, the main effects on Pomc and Htr2c expression are from antibiotic treatment. Regardless of the lack of chemotherapy effect, we found it important to include these results so that we could provide a complete discussion of alteration of metabolic pathways.

6. Need to compare the results presented here with previously published results showing that paclitaxel treatment did not alter locomotion (for example, Nguyen et al, Molecular Neurodegeneration, 2021).

We have included the suggested reference and one other that did not demonstrate a reduction in locomotor activity with paclitaxel treatment, however, we are careful to point out that the dose of paclitaxel used in these references are significantly lower that the 30 mg/kg dose that we use in this manuscript. The changes are incorporated on beginning on line 174.

“This is also consistent with the data of others demonstrating a reduction in dark phase horizontal locomotor activity and wheel running.[21] However, some groups have not observed a reduction in locomotion in mice treated with paclitaxel, [22,23] likely due to differences between our high-dose of paclitaxel and treatment with a much lower dose (2 or 10 mg/kg).”

7. The present study only examined the effects that persisted 8 days after completion of treatment. Many side effects of chemotherapy persist for prolonged periods. The authors need to include experiments that address the long duration of the side effects. Does the proposed protection persist after terminating chemotherapy only when antibiotic treatment is maintained or does protection persist when both chemotherapy and antibiotic treatment terminate at the same time?

The authors agree that there is a need in this field to identify the temporal effects of antibiotic treatment on chemotherapy-associated fatigue. However, these studies were beyond the scope and budget of the current study. 

8. Does the proposed protection alter the ability to treat tumor size and/or spread to additional tissues?

The impact of antibiotic treatment on the anti-cancer effects of paclitaxel are not explored here. Prior to moving to moving towards a more clinically-relevant study, it would be essential to determine if there was an interaction between paclitaxel chemotherapy and antibiotic treatment that prevented effective cancer treatment.

9. In the statistical measurements shown on the figures, it is not clear which bars are being compared. Figures should be easily understood, without the need to read the entire text. Please include clarification on the figures.

The authors appreciate Reviewer 3 catching this, the meaning of * and # has been added to Figure legends 2 and 3.

---

## [Decision Letter · Decision Letter 2]

16 Jan 2023

PONE-D-22-14575R2Antibiotic treatment inhibits paclitaxel chemotherapy-induced activity deficits in female micePLOS ONE

Dear Dr. Pyter,

Thank you for submitting your manuscript to PLOS ONE. After careful consideration, we feel that it has merit but does not fully meet PLOS ONE’s publication criteria as it currently stands. Therefore, we invite you to submit a revised version of the manuscript that addresses the points raised during the review process.

 While the concerns of reviewer 1 have been satisfied, reviewer 2 who had major concerns with the study declined to review the revised manuscript. And since the previous editor was unable to handle the revision and I agreed with the concerns of both reviewers and was not convinced that all of the concerns were adequately addressed, a third reviewer was asked to critique the revised manuscript. Reviewer 3 noted many of the same concerns and additional ones. Unfortunately, it appears that only minor modifications have been actually made in the manuscript to address the concerns of Reviewer 3. Thus, I will allow the authors one more opportunity to more adequately address this reviewer's concerns. Specially, many of the arguments made in the response to reviewers regarding the limitations of the study need to be included in the discussion. Since the number of samples is insufficient to adequately address some of the conclusions, the authors need to very clearly note that this is a pilot study up front and need to clearly outline the specific limitations, as pointed out by Reviewer 3. Simply stating that there was insufficient budget is not an adequate response for lack of experimental rigor.

We look forward to receiving your revised manuscript.

Kind regards,

Brenda A Wilson, Ph.D.

Academic Editor

PLOS ONE

Reviewers' comments:

Reviewer's Responses to Questions

**Comments to the Author**

1. If the authors have adequately addressed your comments raised in a previous round of review and you feel that this manuscript is now acceptable for publication, you may indicate that here to bypass the “Comments to the Author” section, enter your conflict of interest statement in the “Confidential to Editor” section, and submit your "Accept" recommendation.

Reviewer #3: (No Response)

2. Is the manuscript technically sound, and do the data support the conclusions?

Reviewer #3: No

3. Has the statistical analysis been performed appropriately and rigorously? 

Reviewer #3: I Don't Know

4. Have the authors made all data underlying the findings in their manuscript fully available?

Reviewer #3: Yes

5. Is the manuscript presented in an intelligible fashion and written in standard English?

Reviewer #3: Yes

6. Review Comments to the Author

Reviewer #3: The authors have made only modest changes to the manuscript. The conclusions must be tempered by the limitations of the study.

It is noted that chemotherapy regimen is now clearly stated throughout. Note also that the marked copy of the manuscript does not match the revision at the beginning of the file.

There must be a section devoted to limitations. This section needs to include:

Only female mice were included in the study.

The number of mice included in each group is very limited.

The measured effects on were small and had a large variance. This includes the effects on ghrelin and leptin. The effect on activity shows a trend, but was not significant.

The effect of antibiotic treatment on the ability to shrink tumors was not assessed.

The effects of antibiotic treatment was not specific for chemotherapy, that the antibiotic treatment altered the gut microbiome regardless of the presence of paclitaxel.

Bacterial levels return by the end of antibiotic treatment. This response may impact other aspects of cancer treatment.

The present study only examined the effects that persisted 8 days after completion of treatment. Further studies are needed because many side effects of chemotherapy persist for prolonged periods (month to years).

As stated previously: regardless of prior reports, verification that the gut microbiome was altered by the antibiotic treatment must be included. The figure included in the response to reviewers needs to be included in the manuscript. This statement needs to be included: “The recovery of overall bacterial levels at the end of the experiment is likely the result of antibiotic-resistant bacterial taxa blooming in the gut as this analysis represents total bacteria, irrespective of relative abundances of specific taxa. Indeed antibiotics reduce diversity of the gut microbial community without wiping out all types of bacteria; some thrive when others are depleted (Lange et al., 2016).”

As previously stated the authors need to compare the results presented here with previously published results showing that paclitaxel treatment did not alter locomotion (for example, Nguyen et al, Molecular Neurodegeneration, 2021). The authors state:

“We have included the suggested reference and one other that did not demonstrate a reduction in locomotor activity with paclitaxel treatment, however, we are careful to point out that the dose of paclitaxel used in these references are significantly lower that the 30 mg/kg dose that we use in this manuscript. The changes are incorporated on beginning on line 174.”

The citations included are not obviously related to the topic (refs 21,22, 23 appear to relate to statistical analysis and the microbiota). Also, the statement that the treatment levels in the current manuscript are much higher than previous work is not a correct statement. Paclitaxel treatment is often at least 20 mg/kg in previous studies.

As for the figures, it is still not clear which data bars are being compared. There are brackets over all bars in many panel, but it is not clear which comparison is related to the p value presented. Figures need to be updated.

There is a new author, but this author has no contribution listed on lines 322-324

7. PLOS authors have the option to publish the peer review history of their article (what does this mean?). If published, this will include your full peer review and any attached files.

Reviewer #3: No

---

## [Author Response · Author response to Decision Letter 2]

2 Mar 2023

The authors would like to thank the Editor and Reviewer for a final chance to make revisions to this manuscript. Importantly, we have now included a full paragraph of limitations and carefully incorporated changes based on the other comments, as detailed below.

Reviewer Comments

1) It is noted that chemotherapy regimen is now clearly stated throughout. Note also that the marked copy of the manuscript does not match the revision at the beginning of the file.

We agree that it is difficult to go back to the original submission and create a marked copy from all 3 rounds of revisions. However, for the marked version here we have included those of revision 2 in red and this third revision in blue and purple. The clean copy was created as the last step of the resubmission process to ensure no changes were made that were not on the marked copy.

2) There must be a section devoted to limitations. This section needs to include:

Only female mice were included in the study.

The number of mice included in each group is very limited.

The measured effects on were small and had a large variance. This includes the effects on ghrelin and leptin. The effect on activity shows a trend, but was not significant.

The effect of antibiotic treatment on the ability to shrink tumors was not assessed.

The effects of antibiotic treatment was not specific for chemotherapy, that the antibiotic treatment altered the gut microbiome regardless of the presence of paclitaxel.

Bacterial levels return by the end of antibiotic treatment. This response may impact other aspects of cancer treatment.

The present study only examined the effects that persisted 8 days after completion of treatment. Further studies are needed because many side effects of chemotherapy persist for prolonged periods (month to years).

In response to this comment, we have added a paragraph in the Discussion dedicated to the limitations listed above. All limitations listed above were addressed and the paragraph beginning on line 231 reads as follows:

The results of this study are modest and therefore should be regarded in the context of other bodies of work. Future studies will need to address these limitations to significantly impact the understanding of how the gut microbiome plays a role in the development of chemotherapy-associated fatigue. This study was conducted only in female mice and with relatively small sample sizes (n=6-8/group), therefore the statistical power was limited and variability of the biology and activity results were significant. While paclitaxel chemotherapy is used in the treatment of cancer, mice in this study were tumor-naive. Including tumor-bearing mice would provide a more complete understanding of fatigue in cancer patients. The antibiotic treatment used in this study targeted both gram-positive and -negative bacteria and was administered for the entirety of the study to modify the gut microbiome. Future manipulation of the antibiotics regimen and schedule could address the potential for antibiotics, used periodically to resolve infection in chemotherapy patients, to modulate chemotherapy-associated fatigue and energy homeostasis. Furthermore, bacterial sequencing was not repeated for these mice as in our previous study which demonstrated both antibiotic and chemotherapy consequences for the gut relative bacterial taxa abundances.[15] Future bacterial sequencing would allow for correlation analyses between locomotion, hypothalamic PCR, hormone plasma concentrations, and specific gut taxa. Lastly, chemotherapy-associated side effects are known to be persistent in some patients, therefore the focus of this study acutely after treatment does not capture the potential effects of antibiotics on long-term side effects of paclitaxel.

3) As stated previously: regardless of prior reports, verification that the gut microbiome was altered by the antibiotic treatment must be included. The figure included in the response to reviewers needs to be included in the manuscript. This statement needs to be included: “The recovery of overall bacterial levels at the end of the experiment is likely the result of antibiotic-resistant bacterial taxa blooming in the gut as this analysis represents total bacteria, irrespective of relative abundances of specific taxa. Indeed antibiotics reduce diversity of the gut microbial community without wiping out all types of bacteria; some thrive when others are depleted (Lange et al., 2016).”

The graph that was original in the response to reviewers is now included as a supplemental figure. We have included the detail of this result on line 79 and line 91 which now read as follows:

(line 79) As expected, this antibiotic cocktail in chow caused a significant decrease in fecal 16S rRNA as verified in vehicle-treated mice (Supplemental Fig1).

(line 91) By the end of the paclitaxel recovery period, total fecal bacterial DNA (16S rRNA) returned to pre-antibiotic levels despite continuous antibiotic treatment (Supplemental Fig1).

In the discussion we have included this on line 174 which now reads “Further demonstrated here is a recovery of the levels of fecal bacterial DNA (16S rRNA) by the end of the experiment despite continuous antibiotic treatment. This is consistent with the current knowledge that following an initial reduction in bacterial load, resilient bacterial communities will repopulate the gut community.[20]”

4) As previously stated the authors need to compare the results presented here with previously published results showing that paclitaxel treatment did not alter locomotion (for example, Nguyen et al, Molecular Neurodegeneration, 2021). The authors state:

“We have included the suggested reference and one other that did not demonstrate a reduction in locomotor activity with paclitaxel treatment, however, we are careful to point out that the dose of paclitaxel used in these references are significantly lower that the 30 mg/kg dose that we use in this manuscript. The changes are incorporated on beginning on line 174.”

The citations included are not obviously related to the topic (refs 21,22, 23 appear to relate to statistical analysis and the microbiota). 

We appreciate the that the reviewer identified this error with references. There was a mix up in the order of references and a couple seemed to be missing. The references have been corrected to their proper order and now include the references intended to address this comment in the prior resubmission.

21. Ray MA, Trammell RA, Verhulst S, Ran S, Toth LA. Development of a mouse model for assessing fatigue during chemotherapy. 2011; 61:2

22. Toma W, Kyte SL, Bagdas D, Alkhlaif Y, Alsharari SD, Lichtman AH, et al. Effects of paclitaxel on the development of neuropathy and affective behaviors in the mouse. Neuropharmacology. 2017;117: 305–15. doi:10.1016/J.NEUROPHARM.2017.02.020

23. Nguyen LD, Fischer TT, and Erlich BE. Pharmacological rescue of cognitive function in a mouse model of chemobrain. Mol Neurodegener. 2021;16:41. doi.org/10.1186/s13024-021-00463-2

5) Also, the statement that the treatment levels in the current manuscript are much higher than previous work is not a correct statement. Paclitaxel treatment is often at least 20 mg/kg in previous studies.

We have removed the qualifier “much” and instead reported the cumulative dose of our treatment regimen, to capture both dose and frequency, for direct comparison to the cumulative dose in the papers we reference for more detail in how the dosing difference could cause locomotion discrepancies between studies.

Line 185 now reads as follows: However, some groups have not observed a reduction in locomotion in mice treated with paclitaxel, [22,23] likely due to differences between our higher-dose paclitaxel (cumulative 180 mg/kg) and those with a lower dose of paclitaxel (cumulative 4 or 80 mg/kg, respectively).

6) As for the figures, it is still not clear which data bars are being compared. There are brackets over all bars in many panel, but it is not clear which comparison is related to the p value presented. Figures need to be updated.

We apologize for the confusion. The p-values listed above bar graphs are the ANOVA main effect results, therefore they do not relate to any pairwise comparison. Indication that these are main effect p-values has been added in the figure legends. This is also the case for brackets above the two portions of graphs in figure 1. Again, it is now indicated in the figure legend that these p-values are those of the ANOVA main effects.

Figure legend 1, line 98 now reads: The average food intake per mouse per 24-hour period. Each point reflects the previous 48 hr. n=5-8. p-values are representative of ANOVA main effects over the indicated time period.

Figure legend 2 and 3 now both include the statement “p-values above brackets are ANOVA main effect results”

7) There is a new author, but this author has no contribution listed on lines 322-324

We thank the Reviewer for catching this detail. The new authors contributions have been added.

Line 352 now reads as follows: Conceptualization (L.P., K.J.), data curation (K.J., M.S.), data analysis (K.J., C.G., M.S.), funding acquisition (L.P.), manuscript preparation (C.G.). All authors reviewed, edited, and approved the final draft of the manuscript.

Editor Comments

1) Many of the arguments made in the response to reviewers regarding the limitations of the study need to be included in the discussion. 

As indicated in Reviewer comment #2, there is now a complete limitations paragraph included in the discussion to more completely address concerns including sample size and bacterial sequencing.

2) Since the number of samples is insufficient to adequately address some of the conclusions, the authors need to very clearly note that this is a pilot study up front and need to clearly outline the specific limitations, as pointed out by Reviewer 3. Simply stating that there was insufficient budget is not an adequate response for lack of experimental rigor.

It is now made more clear in the Conclusions that this study needs to be followed up with future work. Line 254 now reads “Given the modest sample size (n=6-8), this work may be regarded as preliminary data for future studies. Replication of these findings, further investigation of the effects of antibiotics on reward motivation and effort aspects of fatigue, and studying fatigue caused by other types of commonly used chemotherapeutics will all be important questions for future work.”

See response to Reviewer comment #2 for how we addressed limitations.

---

## [Editor Report · Decision Letter 3]

29 Mar 2023

Antibiotic treatment inhibits paclitaxel chemotherapy-induced activity deficits in female mice

PONE-D-22-14575R3

Dear Dr. Pyter,

We’re pleased to inform you that your manuscript has been judged scientifically suitable for publication and will be formally accepted for publication once it meets all outstanding technical requirements.

Kind regards,

Brenda A Wilson, Ph.D.

Academic Editor

PLOS ONE
---

## [Editor Report · Acceptance letter]

3 May 2023

PONE-D-22-14575R3 

Antibiotic treatment inhibits paclitaxel chemotherapy-induced activity deficits in female mice 

Dear Dr. Pyter:

I'm pleased to inform you that your manuscript has been deemed suitable for publication in PLOS ONE. Congratulations! Your manuscript is now with our production department. 

Kind regards, 

on behalf of

Dr. Brenda A Wilson 

Academic Editor

PLOS ONE